# Simulation of Tetrahedral Profiled Carbon Rovings for Concrete Reinforcements

**DOI:** 10.3390/ma16072767

**Published:** 2023-03-30

**Authors:** Paul Penzel, Tobias Georg Lang, Philipp Benjamin Weigel, Thomas Gereke, Lars Hahn, Arthur Hilbig, Chokri Cherif

**Affiliations:** 1Institute of Textile Machinery and High Performance Material Technology (ITM), Technische Universität Dresden, 01069 Dresden, Germany; 2Institute of Machine Elements and Machine Design, Technische Universität Dresden, 01069 Dresden, Germany

**Keywords:** carbon-reinforced concrete, bond behavior, bond simulation, profiled rovings, pullout test, 3D scan

## Abstract

Textile reinforcements are increasingly establishing their position in the construction industry due to their high tensile properties and corrosion resistance for concrete applications. In contrast to ribbed monolithic steel bars with a defined form-fit effect, the conventional carbon rovings’ bond force is transmitted primarily by an adhesive bond (material fit) between the textile surface and the surrounding concrete matrix. As a result, relatively large bonding lengths are required to transmit bond forces, resulting in inefficient material utilization. Novel solutions such as tetrahedral profiled rovings promise significant improvements in the bonding behavior of textile reinforcements by creating an additional mechanical interlock with the concrete matrix while maintaining the high tensile properties of carbon fibers. Therefore, simulative investigations of tensile and bond behavior have been conducted to increase the transmittable bond force and bond stiffness of profiled rovings through a defined roving geometry. Geometric and material models were thus hereby developed, and tensile and pullout tests were simulated. The results of the simulations and characterizations could enable the optimization of the geometric parameters of tetrahedral profiled rovings to achieve better bond and tensile properties and provide basic principles for the simulative modeling of profiled textile reinforcements.

## 1. Introduction

In recent years, the textile concrete sector has increasingly established itself as an innovation driver in both the construction and textile industries [1]. According to the forecasts of leading German research institutes, around 20% of today’s reinforced concrete will be replaced by carbon concrete within the next ten years [2]. Since the start of research on textile-reinforced concrete in the mid-1990s [3], its development has been pushed forward continuously and intensively [4,5], and rapid transfer to practice has been ensured through close cooperation with companies. Interest is growing steadily and networks for textile-reinforced concrete, such as Composites United-Bau e.V. and Carbon Concrete Composite (C³) e.V., belong to over 122 companies. The primary areas of application are plate- and shell-like constructions [6,7], so flat structures made of carbon fibers with a primarily biaxial structure are of particular relevance.

The bond between rovings and concrete is—similarly to steel-reinforced concrete—mainly based on three mechanisms: adhesion, mechanical interlock and friction [8,9,10].

Due to the high load-bearing behavior and small cross-sections required for textile-reinforced concrete, the bond between the textile reinforcement and concrete matrix is of great importance. In order to effectively transfer the forces between concrete and textile, a permanently rigid and non-slip bond is essential, which was severely limited using previous impregnation technologies. So far, the bonding forces have only been transmitted via material-to-material solutions based on the impregnation agents used.

These solutions have been extensively examined and further developed through numerous research projects and the limits of this technology have been worked out, for example, within the SFB 528 project [11] and the BMBF project C³ [12]. With the latest impregnation systems, these limits have a transferable shear flow of up to 40 N/mm. However, the high impregnating agent content of around 30% by volume results in very stiff structures. In addition, the bond parameters are still insufficient and require overlapping lengths of up to 1 m. The results of national and international research groups, as well as our own preparatory work, show that the profiling of textile reinforcements is essential for a sufficient mechanical bond and load transmission between textile and concrete [12,13,14,15,16].

The modification of the yarn structure necessary for profiling influences yarn characteristics, particularly in terms of the stiffness and strength properties of the yarn. Due to deviation or interruption in a linear filament course, tensile properties are reduced by using common yarn profiling techniques. Moreover, for effective use in concrete structures, textile reinforcement must absorb the tensile forces to be dissipated even with expansions of around 0.1–0.2%. Known processes for generating a profiled yarn surface are cabling, twisting, conventional braiding or the Kemafil process. In all cases, the principle is the formation of a defined yarn geometry by alternately inserting, intertwining or winding the individual threads. As a result, this leads on the one hand to an improvement in the fiber structure inside the yarn and on the other hand, to an improved matrix connection on the yarn surface [17]. However, the principle-related disadvantage of all of the above methods is the associated structural expansion when a load is applied for the first time, which is often in the order of at least 1% elongation and thus represents a clear exclusion criterion for use as concrete reinforcement. In addition, without special consolidation measures, such profiled yarns usually have an insufficient dimensional stability in their cross-section. Especially in the case of core-sheath structures, such as in conventional braiding or Kemafil technology, this causes the shear structure to slide open and thus leads to a telescoping structural failure in the concrete bond, which usually occurs early on.

An alternative is a method in which, after yarn and/or textile fabric formation, contour-shaped fiber polymer composites are produced in a downstream processing step using polymeric matrix material in combination with thermal processes in the pultrusion process [18]. Strong and rigid rods are created, which achieve high bond values by the profiled geometrics, but can no longer be processed in terms of their textile technology.

For the development of carbon-reinforced concrete (CRC) structures with controllable and predictable bond behavior based on a defined form-fit effect, profiled rovings with a patented tetrahedral geometry were developed [19]. Therefore, a new profiling unit for a continuous production of tetrahedral profiled rovings was developed. The tetrahedral roving geometry is illustrated in Figure 1.

A detailed presentation of the profiling unit as well as the general production process and the tetrahedral roving geometry is given in [20,21].

Preliminary investigations have shown that due to their profiling technology and roving geometry, tetrahedral profiled rovings transmit up to 500% higher bond forces [20,21] compared to impregnated rovings without profiling while maintaining high tensile properties. An extensive presentation of the profiling technology and roving properties is given in [20,21].

The numerical modeling and simulation of the structural–mechanical properties of textile structures based on the finite element method (FEM) enables their computer-aided design and design with significantly reduced experimental effort. However, the prediction quality of an FEM analysis of a textile structure depends heavily on the model, i.e., its geometry, the material model and the associated boundary conditions as well as the contact conditions. For this purpose, a large number of models for homogeneous textile structures has been successfully developed and validated. With macroscopic approaches, the textile is modeled as a material with homogenized properties [22,23,24,25]. Discrete approaches are based on the representation by yarns, individual fibers or filaments [26,27,28,29,30,31].

To realize composite-optimized profiled reinforcement yarns, a simulation model for the geometric and material description of tetrahedral profiled rovings was developed in this study. It enables the prediction of the structural–mechanical and composite-side properties in simulated yarn pull and concrete pullout tests. Simulation allows for the selection and further development of promising roving configurations. In contrast to profile-free rovings, tetrahedral profile rovings have an inhomogeneous structure, which must be taken into account when modeling their geometric and material properties. This paper presents simulative approaches for the generation of realistic geometric and material behavior of tetrahedral profiled rovings.

## 2. Materials and Methods

### 2.1. Rovings with Different Configurations

A carbon fiber heavy tow (CFHT) from Teijin Carbon Europe GmbH (Wuppertal, Germany) called Teijin Tenax-E STS 40 F13 48 K 3200 tex was selected to investigate the influence of profiling on mechanical properties. All of the different rovings in this study were produced with this CFHT. Tensile strength was determined in single yarn tensile tests according to the ISO 3341 [32,33]. The manufacturers’ specifications are given in [34].

The rovings were impregnated using a polymeric dispersion on an acrylate base, i.e., TECOSIT CC 1000 from CHT Germany GmbH (Tübingen, Germany) as an impregnation agent. The properties of the impregnation agent are given in [21].

The mass content of the impregnation agent (polymer) of the impregnated rovings, $M_{impregnated}$ (in mass %), was determined by a weight comparison of dry and impregnated rovings according to
(1)$$M_{impregnated}=\frac{m_{impregnated}-m_{dry}}{m_{impregnated}}\cdot 100\%$$
where $m_{dry}$ is the mass of the dry roving and $m_{impregnated}$ is the mass of the impregnated roving (in g).

The fiber–volume content of the impregnated rovings $V_{roving}$ (in vol. %) was determined by transforming the mass content of the impregnated rovings into a volume considering the density of single roving components as follows:(2)$$V_{roving}=\frac{m_{dry}\cdot \rho_{fiber-}m_{impregnated}\cdot \rho_{polymer}}{m_{impregnated}}\cdot 100\%$$
where $\rho_{fiber}$ is the density of the fiber of the dry roving and $\rho_{polymer}$ is the density of the polymer of the impregnation agent (in g/cm³).

For the development of the material and geometric model of the profiled rovings, impregnated rovings with different mass contents and, therefore, different fiber–volume contents and cross-sections, as well as tetrahedral profiled rovings with different profile configurations, were produced on a continuous working laboratory profiling unit. The characteristics of the different rovings are detailed in Table 1.

The profile of the tetrahedral-shaped rovings is characterized by the difference between the minimum and maximum diameter in a profile indentation (smallest cross-section) and the angle of the filament orientation. The angle α is determined as the tangent between the distance of two neighboring profile indentations (profile spacing *t* in mm) of the vertical and horizontal plane (set to 10 mm) and the difference between the minimum and maximum diameter (*d_min_* and *d_max_* in Figure 1), *d_diff_*, as follows:(3)α=arctan⁡ddifft2

An impregnated roving with no profile has a circular shape. Due to the different mass contents and, therefore, different fiber–volume contents, the diameters of the impregnated rovings without profiles varied between 1.8 mm and 2.4 mm.

To investigate the influence of the fiber–volume content, the consolidation parameters and profiling on tensile properties for the development of a material model, different test specimens of impregnated rovings with and without profiling were produced on the laboratory unit (see Table 2). Therefore, impregnated rovings without profiles and different fiber–volume contents were produced using different squeezing devices made of silicon-defined circular punctures (1.5 mm/2.0 mm/2.5 mm/3.0 mm). To investigate the influence of the consolidation parameters, profiled rovings with intensified consolidation were produced by reducing the production speed and therefore increasing the consolidation time using IR radiation according to the setup described in [21]. The tetrahedral profile varied (light or strong) by changing the vertical distance between the upper and lower profiling chain.

### 2.2. Concrete Matrix

Fiber-based reinforcements are often embedded in cementitious matrices with a small maximum grain size, also called mortars [1,35]. For such fine concrete matrices, compressive strength and flexural tensile strength are determined according to the DIN EN 196-1 [36] after 28 days.

For the investigation of bond behavior, a fine concrete dry mix called BMK 45-220-2 was used for the pullout tests at the Institute of Construction Materials (IfB) of TU Dresden. The detailed composition and properties of the concrete are presented in [21].

### 2.3. Test Program and Test Setups

To analyze the roving geometry, the impregnated 100 mm long rovings with and without tetrahedral profiles were digitized by an optical measuring machine using the GOM ScanCobot in connection with the structured-light 3D scanner ATOS Q 12M MV100 from Carl Zeiss GOM Metrology GmbH (Braunschweig, Germany). This measurement setup allowed for the accurate and repeatable acquisition of a measurement uncertainty of 1 µm, a maximal divergence of 20 µm and a minimum measuring point distance of 29 µm per single capturing view. All capturing views were automatically registered through 0.8 mm reference points, while the rovings were surface-treated by an air-dissolving scanning spray as a matting agent to cover up reflective and transparent surface parts. For the exact evaluation of the roving geometry, the software Geomagic Wrap from 3D Systems, Inc. (Rock Hill, SC, USA) and ANSYS SpaceClaim from ANSYS, Inc. (Canonsburg, PA, USA) were used.

The tensile tests were conducted based on the DIN EN ISO 10618 [32] (see also [33,36]). The test setup and testing parameters are described in [20,21].

The characterization of the bond between textile reinforcement and concrete is possible in different ways. However, there is no standardized test method yet. Therefore, a preliminary test method suitable for single yarn pullout tests was examined, mainly to understand whether this test method was suitable for investigating profiled yarns.

Single yarn pullout (YPO) tests were conducted at the Institute of Construction Materials (IfB) at TU Dresden to analyze the characteristic bond–slip behavior of single rovings with different profile properties (see, e.g., [37]). In this type of experiment, individual rovings were embedded in cubic concrete blocks. The upper block provided an embedment length of 50 mm in the top roving section. The lower block possessed an increased embedment length of 90 mm in the bottom roving section for a defined roving fixation. The concrete cover was 40 mm. The specimens were fixed to an upper and lower specimen holder and the pullout force–slip deformation curve was measured by a single-sided pullout in the upper concrete block with a controlled quasi-static load. The pullout (slip) deformation was measured by an optical system consisting of laser sensors and aluminum clips that were fixed to the yarn. The detailed test setup is presented in [21].

### 2.4. Specimen Manufacture

For microscopic examinations, resonated short roving sections of about 10 mm were placed in cylinders of 20 mm in diameter. After one day of drying, the front side was grinded with sandpaper and finally polished.

For the yarn tensile tests, 450 mm long rovings were cast on each end in a 125 mm long epoxy resin embedment. Further details are given in [21].

Specimens for the YPO tests were made by embedding single profiled carbon rovings as well as rovings with no profile in the self-compacting fine-grained concrete BMK 45-220-2 in a cube formwork according to [20,21].

### 2.5. Modeling and Simulation of Tetrahedral Profiled Rovings

A geometric model and a material model were developed to simulate the tensile and bond tests of the tetrahedral profiled rovings. The challenge was the realistic representation of the complex structure of the roving, taking into account the geometric parameters and the type of impregnation and consolidation. A parametric geometric model was therefore derived from 3D scans of real profiled rovings.

Due to the profiling and changing roving cross-section with slight deviations in its linear filament orientation, an inhomogeneous material model was implemented. Therefore, a correlation between the roving area with respect to fiber–volume content and the tensile properties was derived by analyzing and testing impregnated rovings with different fiber–volume contents.

#### 2.5.1. Geometric Model of Tetrahedral Profiled Rovings

In a first step, 3D scans of real impregnated rovings without a profile and tetrahedral profiled rovings were made. Exemplary illustrations of the different roving configurations are shown in Table 3.

Thus, the strong profiled roving shows a dominant tetrahedral geometry, whereas the impregnated roving without a profile shows a circular cross-section and a smooth surface structure. In order to validate the quality of the impregnation as well as the roving cross-section and roving area, microscopic analyses of different roving sections were made. For this purpose, cross- and longitudinal sections in profile and transition segments between neighboring profile indentations were made according to Figure 2.

The comparison of the cross-sections of tetrahedral profiled rovings is illustrated in Table 4. There is a distinctive difference between the roving cross-sections of the profile segments of light and strong profiled rovings. The strong profiled rovings show a dominant rectangular cross-section due to the profiling, whereas the light profiled rovings show a more quadratic cross-section. The strong profiled rovings have a more significant difference between their minimum and maximum diameter (~1.0 mm), and according to Equation (3), a greater angle α (~5°). Both profiled rovings show a circular cross-section in the transition area between the neighboring profile segments. The longitudinal cross-section illustrates the transition and changing diameter along the transition between the horizontal and vertical profiles of the rovings. The different components of the microsection analysis (filament—white, polymer—light grey, embedment resin—dark grey, air void—black) are highlighted in the longitudinal sections.

For the purpose of modeling the yarn geometry, a parametric geometry model was determined, in which the tetrahedral geometry could be adapted by defined parameters. To approximate the geometric model parametrization by the 3D scans, a superelliptical function [38] was chosen as a base for the surface modeling of the periodic geometric model as follows:(4)xs,tys,tzs,t=±a2⋅cos⁡t2n±b2⋅sin⁡t2ns for 0≤s≤zper; 0≤t≤π2

Therefore, a and b describe the maximum axis dimensions of the superellipse in the x- and y-directions, respectively, while the exponent n influences the general shape. By varying the exponent n, the rectangular cross-sections of the tetrahedral profiled yarns can be reproduced as can be seen in Figure 3.

To consider a profile progression in dimension with the variable s, trigonometrically based functions were selected for a and b accordingly:(5)$$\mathrm{Profile}\;\mathrm{dimension}\;\mathrm{along}\;\mathrm{y-axis}:\;a(s)=a_{m}+\sin\left(\frac{2\pi \cdot s}{a_{p}}+a_{o}\right)\cdot a_{h}$$
(6)$$\mathrm{Profile}\;\mathrm{dimension}\;\mathrm{along}\;\mathrm{x-axis}:\;b(s)=b_{m}+cos\left(\frac{2\pi \cdot s}{b_{p}}+b_{o}\right){\cdot a}_{h}$$

Therefore, the parameter $a_{m}$ describes the offset, $a_{h}$, the amplitude, $a_{p}$, the period and $a_{o}$, the phase shift of the function (without units). The parameters $b_{m},b_{h},b_{p}$ and $b_{o}$ are used analogously for the profile dimensions along the *x*-axis. In order to determine the variation in the profile progression over a period length, the parameters were fitted with the measured 3D scans. The yarn profiles were determined from equidistant cross-sections (distance 0.5 mm) along the yarn axis. For each cross-section, a minimum bounding box was computed to determine the yarn profile in the *x*- and *y*-directions. Equations (5) and (6) were adjusted to the consecutive profile dimensions using a non-linear least squares optimization. The exponent n was determined by adapting the superelliptical function to contours of multiple cross-sections. The optimizations were carried out using the toolbox available in scipy [39]. Finally, the geometric surface was modeled by using Equation (4) (see Figure 4a) and meshed with hexahedral elements (Figure 4b) for further use in LS-DYNA.

#### 2.5.2. Material Model of Tetrahedral Profiled Rovings

Due to the profile-dependent and thus changing geometric and material properties along the yarn axis, an inhomogeneous material model for the profile yarns with varying material parameters along the yarn axes was created. The material parameters are based on roving tensile tests of impregnated rovings with varied yarn cross-sections and fiber–volume contents, to approximate the profile-dependent changes in each cross-section. In Table 5, the microscopic analysis of impregnated rovings with different fiber–volume contents is compared. For this purpose, the impregnated rovings were produced with different punctured silicon squeezers (squeezing gap 1.5/2.0/2.5/3.0 mm), defining the fiber–volume content of the rovings.

Due to the defined squeezing of the impregnated rovings, the fiber–volume contents of the rovings and, respectively, the roving areas varied. The results presented in Table 5 are mean values of five test specimens together with exemplary illustrations. As expected, the fiber–volume content decreased and, respectively, the roving area increased with greater squeezing gaps from approximately 78 vol. % and 2.3 mm² (squeezing gap 1.5 mm) to 44 vol. % and 4.1 mm² (3.0 mm). The impregnated rovings with 1.5 mm and 2.0 mm squeezing gaps thereby show a relatively circular cross-section with a dense fiber arrangement, whereas the impregnated rovings with a 3.0 mm gap show a deformed cross-section and relatively loose fiber arrangement with great polymeric accumulation.

In order to derive a material model that considers the dependency of tensile properties and fiber–volume content, the tensile properties of the impregnated rovings with different fiber–volume contents were determined. Figure 5 shows the resulting correlation between Young’s modulus based on the roving area and the fiber–volume content of the dry and impregnated carbon rovings (dots) with a single standard deviation (error bar) and a linear trend line (dotted line).

Dry rovings had the highest Young’s modulus of 235 GPa. They furthermore had, at 100% fiber–volume content, a dry filament area of about 1.81 mm² and therefore the smallest area resulting in the highest Young’s modulus based on the roving area. With decreasing fiber–volume content, the roving area increases, resulting in a reduced Young’s modulus. Therefore, a linear correlation could be approximated between the data points, resulting in the following function:(7)$$E_{fiber-volume-content}\approx 2.15*V_{roving}+15.36$$

This correlation was considered in the developed material model of the tetrahedral profiled rovings by cutting the profiled rovings into 0.1 mm long segments along the roving axis (see Figure 6a), determining the averaged roving area and resulting fiber–volume content in each segment and assigning each segment to an averaged fiber–volume content based on Young’s modulus according to the trend line in Figure 5.

The deviation in the filaments from the linear course due to the tetrahedral geometry was taken into account by corresponding directional vectors in the material model. The required vector field was generated from the geometric description in Equation (4) (see Figure 6b). The tensile properties of each roving segment were correlated with the corresponding vectors.

The tensile tests of the tetrahedral profile rovings were simulated in LS-Dyna. In accordance with the tensile tests, the rovings were fixed at one end and a displacement in the $u_{z}$-direction was applied to the other end. Figure 7 shows the preset boundary conditions for the virtual tensile test.

#### 2.5.3. Modeling of the Bond Behavior of Tetrahedral Profiled Rovings

The created FE models of the tetrahedral profiled rovings were transferred to a simulation model of the concrete pullout test. Figure 8a shows an example of the modeled pullout test setup.

To mesh the concrete volume, fully integrated hexahedral elements were chosen. The mesh between the roving and concrete was refined with a 3 mm mesh size (Figure 8c). Since the focus was not on modeling the concrete matrix, an existing, simplified material approach was used according to [40,41] with the material parameters presented in Table 6.

Therefore, the parameter $\rho_{concrete}$ describes the density and $f_{c}^{'}$, the negative of the unconfined compression strength. The parameters RSIZE and UCF are needed for the conversion from imperial units to SI units. The parameter $\varepsilon_{max}$ specifies the maximum principal strain and was added in addition to the previously described material model as a variable for element erosion.

The concrete body consists of a rectangle with an edge length of 80 mm in length and width and a height of 50 mm, in the middle of which the geometry of the profiled roving was cut out, and the profiled roving was set into the mold for the bond simulation. The contact area is the surface between the profiled roving and the negative in the concrete mold. The geometry of the profiled roving creates a form-fit-based bond. The mesh of the two surfaces is designed such that the nodes of the finite element mesh of the profiled rovings are congruent with the nodes of the concrete body mesh.

In LS-DYNA, the contact was modeled using the *CONTACT_SURFACE_TO_SURFACE keyword [42]. This model for describing the contact is a two-way contact, i.e., the contact is tested for penetration in both directions, both on the surface of the concrete body and on the surface of the profiled roving.

The model is also based on a penalty contact algorithm. For example, if a node of the profiled roving penetrated the surface of the concrete body, the penetration depth *DP* was calculated. A force *F_N_* was applied to the node. This penalty force is calculated as follows:(8)$$F_{N}=k\cdot DP$$

Hereby, stiffness was also considered. The stiffness constant k is
(9)$$k=\frac{f_{s}\cdot A_{K}\cdot K}{V}$$
where $f_{s}$ is the static penalty factor, $A_{K}$ is the contact area and *K* is the compressive modulus of the material with the lowest stiffness. Therefore, K is calculated according to
(10)$$K=\frac{E}{3(1-2\upsilon )}$$
when the penalty force was applied, the node was projected back onto the surface of the concrete body. The force balance between the nodes of the profiled roving and the concrete body resulted in a force that acted in the opposite direction of the penalty force on the concrete body, which absorbed the force. In addition, the node was assigned a friction force as a product of $F_{N}$ and the material dependent friction coefficient µ (without units) according to
(11)$$F_{F}=µ\cdot F_{N}$$

At this point, the friction coefficient was set to 0.5 on the bases of [43,44].

The friction force simulates the friction-based bond. However, this contact description body cannot simulate the adhesion between the profiled roving and the concrete.

Figure 9 shows the boundary conditions for the simulated pullout test.

## 3. Results and Discussions

### 3.1. Geometric Validation of Tetrahedral Profiled Roving Model

By following the previously described approach, models were derived from scans of the profiled rovings. Based on the optimization, an optimal set of parameters for the Equations (5) and (6) was found. The resulting parameters are provided in Table 7.

In Figure 10, the fitted curves are compared to the test data by superimposing them. The graph in Figure 10a shows the curves depicting the dimension in the x- and y-directions. The local minima of each measured curve lead to an asymmetric curve trajectory, which therefore could be completely depicted by the trigonometric curves. Since the curves are in overall agreement, the fit was considered sufficient. From Figure 10b, the yarn cross-section perpendicular to the yarn axis is shown.

### 3.2. Simulation of the Tensile Properties of Tetrahedral Profiled Rovings

The simulation of a yarn tension test provided the force–strain behavior and the geometric changes occurring under load. In addition, the influence of the geometric parameters was analyzed using simulations. Figure 11 shows a comparison of the measured and simulated force–strain behavior of the strong profiled rovings.

A good correspondence between the measured and simulated force–strain behavior was achieved. Furthermore, the profiled rovings showed a generally good tensile behavior in the range of the impregnated rovings without a profile (tensile strength approx. 3.100 MPa, Young’s modulus approx. 210 GPa). A comparison of rovings with different profile configurations is shown in the diagrams in Figure 12.

Thus, the profiled rovings showed similar tensile properties to the impregnated rovings without a profile, indicating an even load distribution among the single filaments. The configuration of the profile itself had a minor influence, resulting in about 5% less tensile properties of the strong profiled roving in comparison to the light profiled roving. The intensity of the consolidation had an influence on the tensile properties as well. The long consolidated profiled roving showed a 5% higher tensile strength and Young’s modulus compared to the profiled roving with a short consolidation.

For the increased tensile properties, it was assumed that a more intensive consolidation would result in longer and uniform polymer chains and higher impregnation stiffness, therefore increasing the resistance of the profile against deformation under stress (also called pliability).

The simulation results significantly correlated to the real tensile tests with a 2–5% deviation range. Only Young’s modulus of the profiled rovings was about 10% higher than the real tests. A possible reason for this is the linear approach (see Section 2.5.2) of the fiber–volume-based tensile properties and deviation of the fitted profile geometry, resulting in smaller roving areas and, therefore, higher tensile properties.

### 3.3. Simulation of the Bond Properties of Tetrahedral Profiled Rovings

With the simulated pullout test, the qualitative bond behavior of the different roving variants could be determined. The average measured (five test specimens each) and simulated bond–slip relationships of the profiled rovings with different profile configurations as well as the bond strength at 0.5 mm pullout are shown in Figure 13. The bond strength represents the specific pullout load (in N/mm) and refers to the measured bond force (in N) divided by the bond length (in mm).

It is clearly visible that the profiled rovings transmit much higher pullout loads than the impregnated roving without a profile. The strong profiled roving with a long consolidation achieved a bond strength at about 100 N/mm, an almost 500% higher bond strength than the roving without a profile. The roving with normal consolidation showed a 20% decreased bond behavior at about 80 N/mm. A possible reason for this is the mentioned resistance of the profile against deformation under a load. A higher consolidation increases resistance and allows for the transmission of higher bond forces. The light profiled roving showed a bond strength at about 55 N/mm, with 30% smaller bond properties due to the reduced form-fit effect with the surrounding concrete matrix. Nevertheless, it had 250% higher bond properties than the impregnated roving without a profile.

Furthermore, the dependence of bond properties on profile configuration shows the possibility of a predictable design of bond properties through defined profiling. All rovings showed a similar initial bond stiffness, but in correlation with the maximum bond strength, the linear rise flattened out eventually. For the restriction of the crack openings in the concrete structure, the bond strength at the first 1% pullout, which was 0.5 mm in the case of a 50 mm embedment length, was decisive for the selection and design of the textile reinforcement structure. The profiled rovings thus showed a significant increase compared to the impregnated rovings without profile (see Figure 13b).

In general, the simulated bond test showed a very good correlation with the real bond test. Especially in the initial 0.1–0.8% pullout length, the simulation was very accurate (± 5%). With increasing pullout, the simulated bond test showed a slightly higher bond stiffness. Especially in the case of long consolidated profiled rovings, the bond stiffness was very high, resulting in an early failure. In the case of the profiled rovings with normal consolidation with strong and light profiles, the simulation showed a good correlation until 1.0% pullout, but showed a higher bond strength and stiffness. In the case of the impregnated rovings without profile, a very good correspondence was achieved.

Because only the initial 1.0% pullout is significant, the derived model for the simulated bond test is suitable for well-founded statements and predictions of the bond behavior of profiled rovings. Furthermore, the simulation shows the apparent form-fit effect between the profiled roving and the surrounding concrete matrix, resulting in increased bond properties. This can be clearly illustrated by an exemplary stress distribution between rovings (two profile segments) and the surrounding concrete matrix as is shown in Figure 14.

With the developed FE model, pullout tests with different profile configurations were simulated in order to predict their bond behavior and derive suitable profile geometries. For example, Figure 15 shows the bond behavior of strong tetrahedral profiled rovings with a profile spacing of 10 mm and 7.5 mm between neighboring profile indentations.

Based on the simulated bond test, profile spacing has an evident influence on bond behavior. On the one hand, a reduction of the profile spacing from 10 mm to 7.5 mm increased the simulated bond strength from 100 N/mm to almost 140 N/mm (+40%). A major reason for this is that the 50 mm long embedment length included six instead of four profile indentations, resulting in an increased mechanical interlock. On the other hand, the tensile strength of the profiled rovings with reduced profile spacing increased by 3.800 MPa due to the mentioned correlation between the fiber–volume content and the tensile properties (see Equation (4) and Section 3.1). A smaller profile spacing results in more profile indentations and therefore higher fiber–volume content, leading to improved tensile properties. This correlation has to be validated in further studies on real profiled rovings with reduced profile spacing.

## 4. Conclusions

In summary, the presented tetrahedral profiled carbon rovings transmitted up to 500% higher pullout loads compared to impregnated rovings without a profile. In order to predict the tensile and bond behavior of profiled rovings and enable a requirement-based design of textile reinforcement structures, an FE model was derived. For this purpose, a new geometric and material model was developed for the tetrahedral profiled roving. Based on 3D scans and microscopic analyses, a realistic and parametric geometric model with high accuracy was achieved. Due to the profiling and changing roving cross-section with slight deviations of the linear filament orientation, an inhomogeneous material model was implemented. Therefore, correlations between the roving area and fiber–volume content, respectively, and the tensile properties, were derived.

The simulated tensile tests showed a very good correlation with the measured test results. Nevertheless, the FE model showed slightly higher tensile properties due to the overrepresentation of the high fiber–volume content in the profile indentations.

In the simulated bond test, a clear correlation with the real test could be achieved, especially in the initial and most important pullout of about 1.0%. Increased pullouts showed a slight deviation in bond simulation due to higher stiffness of the roving. Thus, the simulation was suitable for predicting the bond behavior of the profiled rovings for concrete applications.

A further parametric study of bond tests with different roving geometries enabled the prediction of the bond behavior of the profiled rovings with reduced profile spacing based on well-founded statements.

In summary, it can be stated that this developed geometric and material model of tetrahedral profiled rovings is suitable for a simulative design and allows for the prediction of their tensile and bond behavior. Furthermore, there is potential in the further development of roving geometry as well as the profiling process to maximize bond strength and bond stiffness by optimizing geometric parameters and the consolidation process. Therefore, extensive further research is planned to find an optimized roving geometry as well as a method for the targeted adjustment of strength and composite properties through the defined and variable profiling of carbon rovings in addition to a numerical description of their bond behavior.

## Figures and Tables

**Figure 1 materials-16-02767-f001:**
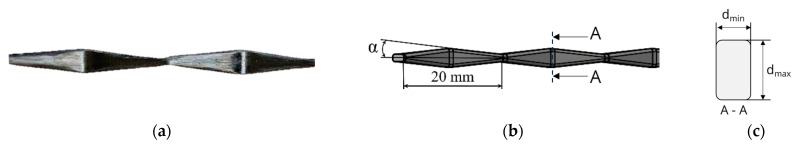
Profiled carbon roving with a tetrahedral geometry: (**a**) photography; (**b**) schematic illustration; (**c**) schematic cross-section (acc. to [20,21]).

**Figure 2 materials-16-02767-f002:**
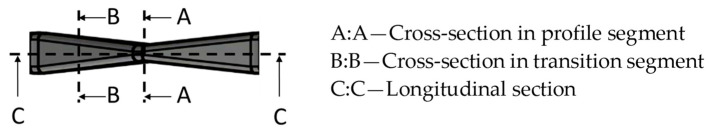
Schematic cross-sections for the microscopic analysis.

**Figure 3 materials-16-02767-f003:**
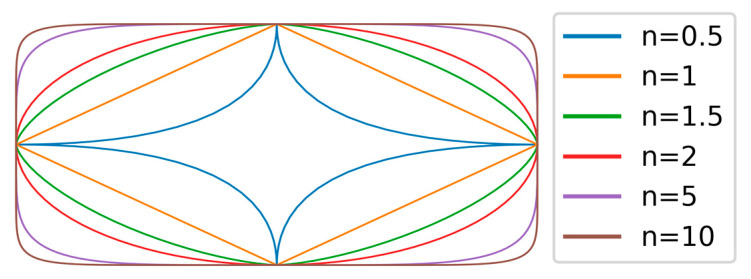
Superelliptical curves with varying curvatures (depicted by exponent n, aspect ratio 2).

**Figure 4 materials-16-02767-f004:**
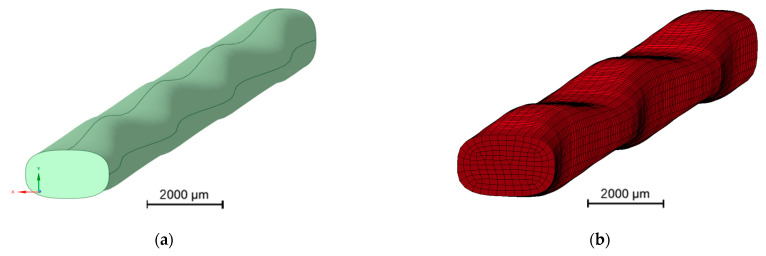
Parametric geometry models of the profiled rovings. (**a**) Parametric surface model, (**b**) Hexahedral mesh of geometry.

**Figure 5 materials-16-02767-f005:**
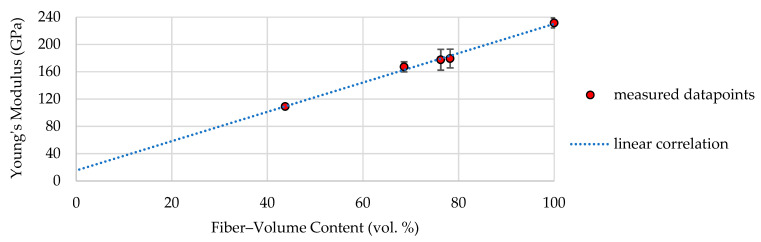
Tensile properties of the dependence of the fiber–volume content of impregnated rovings (3.200 tex with Tecosit as the impregnation agent) tested acc. to the DIN EN ISO 10618.

**Figure 6 materials-16-02767-f006:**
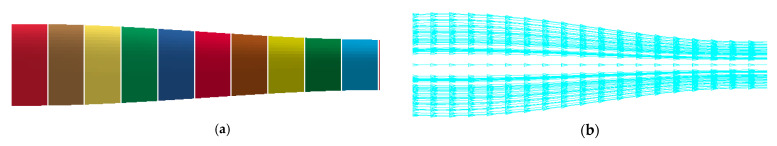
Detail view of yarn model with (**a**) Segmented yarn model, (**b**) Vector field for material orientation definition.

**Figure 7 materials-16-02767-f007:**
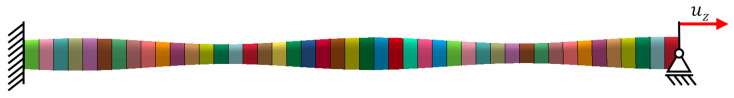
Boundary conditions for simulated tensile test (colors acc. to roving segmentation).

**Figure 8 materials-16-02767-f008:**
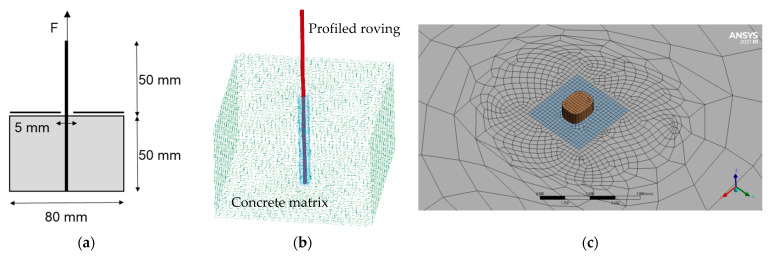
Simulated pullout test setup. (**a**) Schematic setup, (**b**) Meshed model, (**c**) Detail view of model mesh.

**Figure 9 materials-16-02767-f009:**
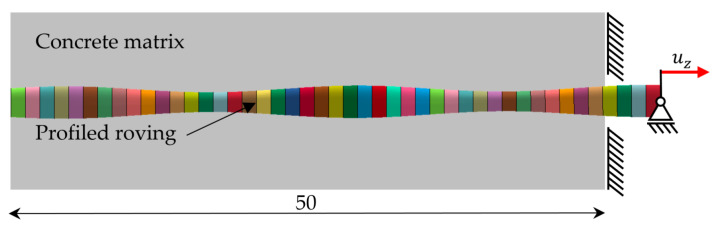
Boundary conditions for simulated pullout test (colors acc. to roving segmentation).

**Figure 10 materials-16-02767-f010:**
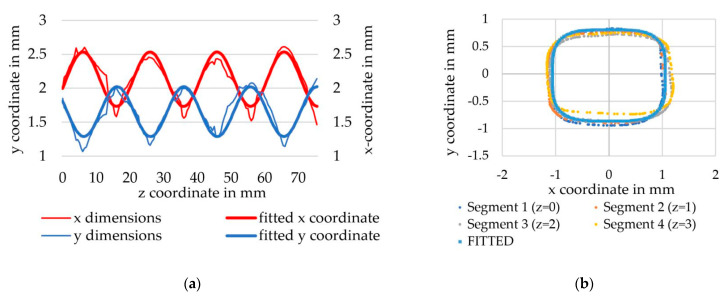
Comparison of derived roving geometry with experimental data. (**a**) Comparison of profile curves, (**b**) Comparison of hull curves.

**Figure 11 materials-16-02767-f011:**
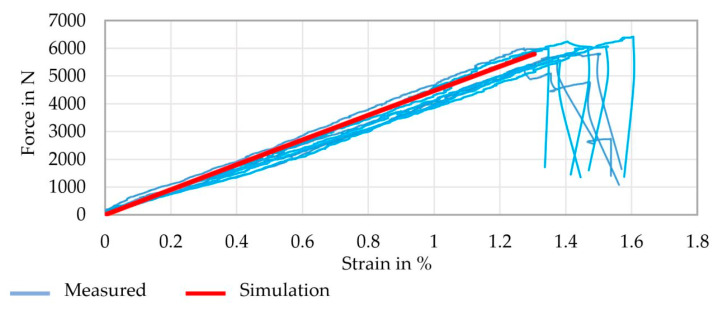
Comparison of the measured and simulated force–strain behavior of profiled rovings with a strong profile.

**Figure 12 materials-16-02767-f012:**
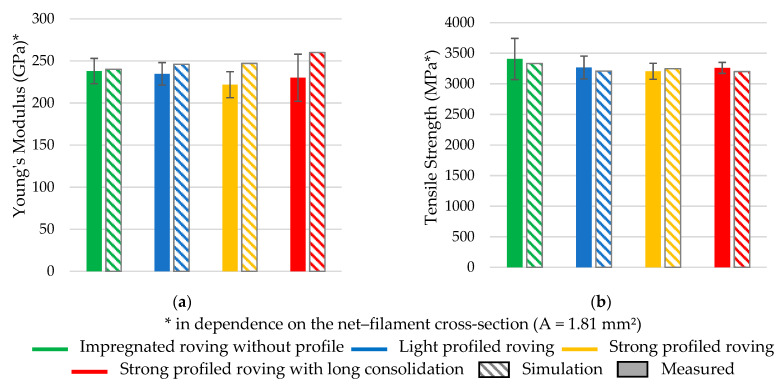
Measured and simulated tensile behavior of different profiled rovings. (**a**) Young’s Modulus, (**b**) Tensile strength.

**Figure 13 materials-16-02767-f013:**
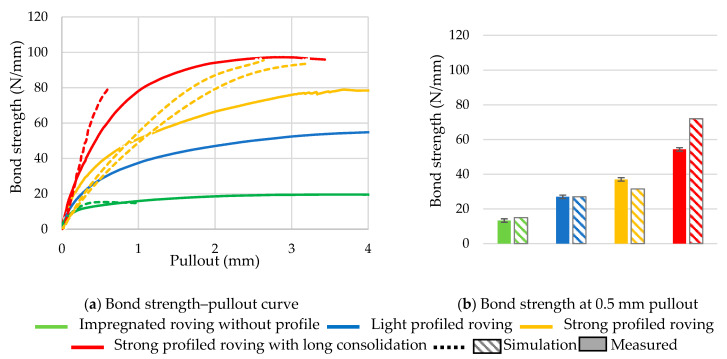
Measured and simulated bond behavior of different profiled rovings. (**a**) Bond-strength–pullout curve, (**b**) Bond strength at 0.5 mm pullout.

**Figure 14 materials-16-02767-f014:**
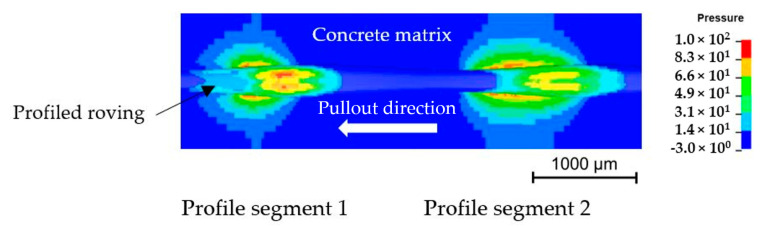
Simulated pressure in a concrete matrix (data in N/mm²).

**Figure 15 materials-16-02767-f015:**
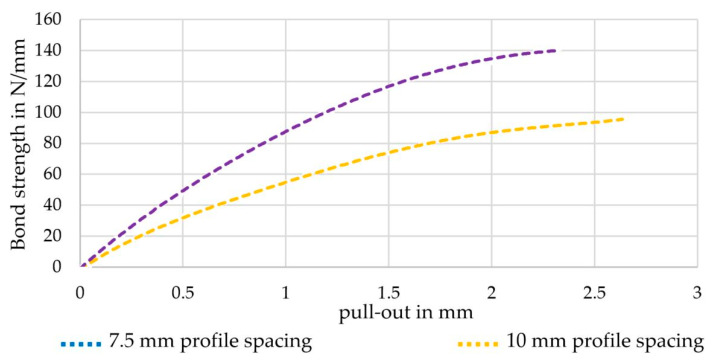
Simulated bond behavior of strong profiled rovings with different profile spacings.

**Table 1 materials-16-02767-t001:** Profile characteristics of different rovings.

Roving Configuration	Roving Geometry	Specimen Illustration	Schematic Cross-Section	Dimension of Cross-Section
Dry yarn	Band-shaped	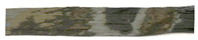		Variable(no internal bond)
Impregnated roving	Circular	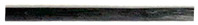	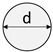	d = 1.8–2.4 mm
Tetrahedral profiledroving	Light profile	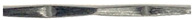	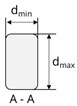	ddiff = 0.8 mmα = 4°
Strong profile	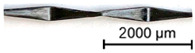	ddiff = 1.0 mmα = 5°

**Table 2 materials-16-02767-t002:** Properties of different rovings.

Roving Configuration	Parameters			
	Roving Geometry	Impregnation Agent	Fiber–Volume Content (%)	Consolidation Time (min)
Rovings without profiles				
Dry roving	Band-shaped	–	100	–
Impregnated roving	Circular tiny	Tecosit	~78	4
	Circular small		~76	
	Circular middle		~69	
	Circular large		~44	
Profiled rovings from laboratory unit with different profiles and impregnation agents				
Tetrahedral profiled roving	Light profile	Tecosit	~74	4
	Strong profile			4
				10

**Table 3 materials-16-02767-t003:** 3D scans of different rovings as the basis for the geometric models.

Roving Geometry	3D Scan
No profile	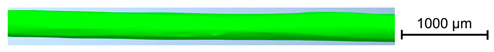
Light profile	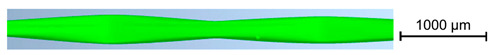
Strong profile	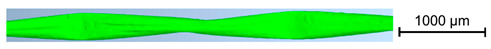

**Table 4 materials-16-02767-t004:** Comparison of the microscopic analysis of different cross-sections of tetrahedral profiled rovings.

Profiled Roving—Cross Section
Profile Segment (Light Profile)	Profile Segment (Strong Profile)	Transition Segment
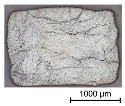	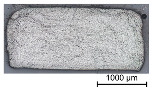	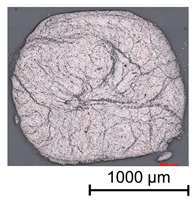
Profiled Roving—Longitudinal Section (profile and transition segment)
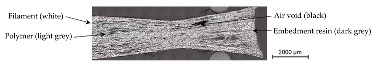

**Table 5 materials-16-02767-t005:** Microscopic analysis of impregnated rovings with different fiber–volume contents.

Squeezing gap: 1.5 mm Fiber–volume content: 78.2% Roving area: 2.31 mm²	Squeezing gap: 2.0 mm Fiber–volume content: 76.2% Roving area: 2.37 mm²
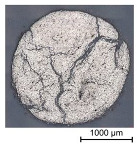	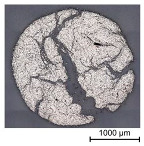
Squeezing gap: 2.5 mm Fiber–volume content: 68.6% Roving area: 2.63 mm²	Squeezing gap: 3.0 mm Fiber–volume content: 43.8% Roving area: 4.1 mm²
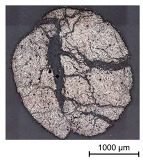	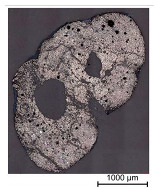

**Table 6 materials-16-02767-t006:** Concrete material parameters for bond simulation.

Parameter	Symbol	Value	Unit
RO	$\rho_{concrete}$	2.26	g/cm³
A0	$f_{c}^{'}$	−101	MPa
RSIZE	–	0.03937	In/mm
UCF	–	145	Psi/MPa
MXEPS	$\varepsilon_{max}$	0.08	

**Table 7 materials-16-02767-t007:** Fitted geometric parameters (see Equations (5) and (6)).

	$a_{m}$	$a_{p}$	$a_{o}$	$a_{h}$	$b_{m}$	$b_{p}$	$b_{o}$	$b_{h}$	n
Unit	mm	mm	1	mm	mm	mm	1	mm	1
Value	1.0563	19.9115	−3.4164	0.1995	0.8387	19.7746	−1.9632	0.3611	3.4301

## Data Availability

The data presented in this study are available upon request from the corresponding author. The data are not publicly available due to privacy concerns.

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
