# Peer review of "Simulation of Tetrahedral Profiled Carbon Rovings for Concrete Reinforcements"

_materials, 2023, doi:10.3390/ma16072767_

Round 1
Reviewer 1 Report
1. The concrete mix should be provided. On the other hand, the maximum grain size is 2 mm, this concrete is probably more accurately called mortar.
2. How about the bonding property between this fiber and normal weight concrete (NWC)? Is it suitable for NWC? Is it primarily used for reinforcement or something?
3. How is this friction coefficient between this fiber and concrete matrix measured? What is its value?
4. In figure 18, there is no descent part of simulation curve, why?
5. Is the unit of bond strength N/mm2 (MPa) or N/mm?
6. There are some mistakes on spelling.
Author Response
Dear reviewer,
thank you very much for your helpfull comments and advises. I implemented your recommendations in the paper as stated in the following points (italic):
- The concrete mix should be provided. On the other hand, the maximum grain size is 2 mm, this concrete is probably more accurately called mortar.
The detailed composition oft he concrete is given in a different paper [21].
„ For the investigation of the bond behavior, a fine concrete dry mix called BMK 45-220-2 was used for the pull-out tests at the Institute of Construction Materials (IfB) of TU Dresden. The detailed composition of the concrete is presented in [21].“
In the following, the fine concrete is refered to as mortar.
- How about the bonding property between this fiber and normal weight concrete (NWC)? Is it suitable for NWC? Is it primarily used for reinforcement or something?
Due to the properties and application of carbon reinforced concrete, the use of fine concerete or mortar has been established. The use of nomal weight concrete is not established due to conventionally small grid of the textile structure (25.4 x 25.4 mm²) and the hempered penetration of big grain sizes above 2 mm. The use of NWC is for such structures unsuitable but will be studied in future researches. The represented textile structures are primarily used for reinforcement of consiting structures and therefor a mortar for sparying or lamination is used.
- How is this friction coefficient between this fiber and concrete matrix measured? What is its value?
On basis of other experimental and numerical studies [43, 44], the friction coefficient was set to 0.5.
- In figure 18, there is no descent part of simulation curve, why?
The simulation stops at yarn/roving failure. Therefor no force drop or descent part of the simulation curve is shown. The descent of the measured data is due to the test setup and stop condition (force drop by 60 % etc.)
- Is the unit of bond strength N/mm2(MPa) or N/mm?
The represented bond strength is similar to the specific pull-out load and refers to the measured bond force (in N) divided by the bond length (in mm) and is given in N/mm. The resulting bond stress would be given in N/mm².
- There are some mistakes on spelling.
Minor typos have been corrected.
Reviewer 2 Report
This work deals with the numerical model of different geometries of tetrahedral profiled rovings embedded in concrete in order to study their influence on the mechanical properties (tensile strength, young modulus and bond strength) of the composite.
In general terms, the manuscript is well written and organized.
The bibliography is consistent with the topic but more references about numerical modelling would be desirable, even to, if possible, be able to compare the obtained results with those of other authors.
In addition, there are not many updated references. In this regard, the reviewer only finds around 15% of the bibliography published between 2018 and 2020. The most updated references correspond to works by the same authors in 2022.
The work is excessively long and repeats content already published in references 20 and 21 by the same authors. This fact could blur the reader. For example, sections 2.1, 2.2, 2.3 and 2.4 should be removed. Also, please avoid copying textually paragraphs from previous publications, such as lines 180 to 183, and lines 188 to 194, which are textually identical to what was written in reference 21. In the aforementioned subsections, the only thing that stands out as original is the text "For the exact evaluation of the roving geometry, the software Geomagic Wrap from 3D Systems, Inc. (USA) and ANSYS SpaceClaim from ANSYS, Inc. (USA) were used."
In the Introduction Figure 1 is irrelevant.
Figures 2, 3, 4, 5, 6, 7 and 8 can be eliminated as they are from reference 21.
Tables 1, 2 and 5 are exactly the same as the tables in reference 21.
Regarding numerical modelling, the results are interesting and well presented. In most cases, the predictions overestimate the actual behavior of the bond strength of the composite. About this fact, the authors expressly clarify that “Because only the initial 1.0 % pullout is significant, the derived model for the simulated bond test is suitable for well-founded statements and predictions of the bond behavior of profiled rovings”. In this regard, the reviewer would like to ask the authors why they think that the predictions show this overestimation that is even greater the higher the strength of the modeled material. Even this overestimation occurs in values less than 1% for the case "Strong profiled roving with long consolidation". What is the authors opinion about this particular case?
Author Response
Dear reviewer,
thank you very much for your helpfull comments and advises. I implemented your recommendations in the paper as stated in the following points (italic):
The bibliography is consistent with the topic but more references about numerical modelling would be desirable, even to, if possible, be able to compare the obtained results with those of other authors.In addition, there are not many updated references. In this regard, the reviewer only finds around 15% of the bibliography published between 2018 and 2020. The most updated references correspond to works by the same authors in 2022.
Recent studies from other researchers have been added, e.g. [43]. A comparison with other authors is difficult, because recent studies have focused on thick FRP-rebars. The profiled rovings are novel and distinguish themselves by a thin diameter in contrast to the profiled rebar structures.
The work is excessively long and repeats content already published in references 20 and 21 by the same authors. This fact could blur the reader. For example, sections 2.1, 2.2, 2.3 and 2.4 should be removed. Also, please avoid copying textually paragraphs from previous publications, such as lines 180 to 183, and lines 188 to 194, which are textually identical to what was written in reference 21. In the aforementioned subsections, the only thing that stands out as original is the text "For the exact evaluation of the roving geometry, the software Geomagic Wrap from 3D Systems, Inc. (USA) and ANSYS SpaceClaim from ANSYS, Inc. (USA) were used."
Identical text passages have been edited or removed. Section 2.1, 2.2, 2.3 and 2.4 have been shortend and referenced to [20, 21].
In the Introduction Figure 1 is irrelevant.
This figure 1 has been removed.
Figures 2, 3, 4, 5, 6, 7 and 8 can be eliminated as they are from reference 21.
Figures 2, 4, 5, 6, 7 and 8 have been deleted and referenced with [21]. Figure 3 is needed for a better understanding oft he roving geometry and geometry configuration.
Tables 1, 2 and 5 are exactly the same as the tables in reference 21.
Tables 1, 2 and 5 have been deleted and referenced with [21].
Regarding numerical modelling, the results are interesting and well presented. In most cases, the predictions overestimate the actual behavior of the bond strength of the composite. About this fact, the authors expressly clarify that “Because only the initial 1.0 % pullout is significant, the derived model for the simulated bond test is suitable for well-founded statements and predictions of the bond behavior of profiled rovings”. In this regard, the reviewer would like to ask the authors why they think that the predictions show this overestimation that is even greater the higher the strength of the modeled material. Even this overestimation occurs in values less than 1% for the case "Strong profiled roving with long consolidation". What is the authors opinion about this particular case?
Due to the chosen scale and idealised assumptions on the concrete geometry, local variations of the concrete and the fibre-concrete interface could not be adequately represented. In recent tests, an increased number of pores could be seen in the fibre-concrete interfaces. These lead to a different contact behaviour and could possibly cause the deviation between experiment and simulation.
The focus of this study is a qualitative comparison of the influence of different profile configurations on the bond behavior in concrete in order to allow predictions and derive favorable geometric parameters. An exact representation of the bond behavior is not the main goal of the presented investigation.
Round 2
Reviewer 2 Report
The reviewer's suggestions and comments were adequately addressed by the authors. For this reason, the reviewer considers that the manuscript should be published in the journal Materials.